# Intralaboratory Validation of a Kinetic Turbidimetric Assay Based on Limulus Amebocyte Lysate (LAL) for Assessing Endotoxin Activity in Cow Milk

**DOI:** 10.3390/ani13030427

**Published:** 2023-01-27

**Authors:** Pablo Flórez, María de Castro, David Rodríguez, José Manuel Gonzalo-Orden, Ana Carvajal

**Affiliations:** 1Laboratorios Analíticos AGROVET, N-601 km312, 24217 Mansilla Mayor, León, Spain; 2Departamento de Medicina y Cirugía Animal, Universidad de León, 24071 León, Spain; 3Instituto de Biomedicina, IBIOMED, Universidad de León, 24071 León, Spain; 4Departamento de Sanidad Animal, Universidad de León, 24071 León, Spain; 5Instituto de Desarrollo Ganadero, INDEGSAL, Universidad de León, 24071 León, Spain

**Keywords:** cattle, endotoxin, limulus amebocyte lysate, milk, mastitis, lipopolysaccharide

## Abstract

**Simple Summary:**

Rapid quantification of endotoxins in raw milk samples would be of interest for differentiating Gram-positive and Gram-negative mastitis and would contribute to their clinical management. In this research, we have validated a kinetic turbidimetric assay based on Limulus amebocyte lysate for endotoxin quantification in milk samples. This assay was demonstrated to be robust and useful in avoiding the effect of colour and interfering substances in milk samples and allowing for the identification of coliform mastitis in milk samples from affected cows.

**Abstract:**

Mastitis, one of the most common diseases in dairy cattle, causes severe losses in the dairy sector worldwide and affects animal welfare. The disease is characterized by an inflammatory reaction of the mammary gland and is mainly caused by bacterial infections, including both Gram-negative and Gram-positive bacteria. The release of endotoxins associated to bacterial lysis is a weighty factor in the clinical course of Gram-negative associated mastitis and should be taken into consideration when using antibiotics in the treatment of these infections. Therefore, endotoxin detection in milk samples would be of help in the management of bovine mastitis. With this aim, we have validated a kinetic turbidimetric assay based on Limulus amebocyte lysate (LAL) for the quantification of endotoxins in milk samples. The assay was adapted to this particular matrix by incorporating filtration and dilution of the milk samples in the procedure. Our results demonstrate the robustness and usefulness of the assay, which allows the identification of coliform mastitis in milk samples from affected cows and the quantification of endotoxin activity in bulk and commercial milk samples. Further studies are required to evaluate the performance of the assay in mastitis milk samples associated to Gram-negative bacteria other than *Escherichia coli* as well as during the clinical course of these Gram-negative mastitis or after their treatment with antibiotics.

## 1. Introduction

Endotoxins are large and heat stable amphiphilic molecules present in the outer membrane of Gram-negative bacteria and consisting of lipopolysaccharides (LPS) (lipid A attached to a carbohydrate core and polysaccharide O). They are released upon bacterial death or during cell division and growth [1] and are good indicators of active infection, being very suitable as markers for its acute phase [2]. Endotoxins interact with membrane receptors of the cells of the immune system, such as neutrophils and macrophages, triggering the production of proinflammatory cytokines as interleukins and tumour necrosis factor alpha and inducing inflammation [3]. High levels of endotoxins may cause a disproportionate innate immune response that may become deleterious to the host [4,5].

Mastitis is one of the most common diseases in dairy cattle. It affects animal welfare and causes severe losses in the dairy sector worldwide due to decreased milk production, poor milk quality and increased culling rates [6]. The disease is mainly caused by bacterial infections, including both Gram-negative and Gram-positive bacteria, and is characterized by an inflammatory reaction of the mammary gland. The release of endotoxins and its translocation from the mammary gland into the systemic circulation contribute to the clinical disease in Gram-negative associated mastitis [6]. Moreover, when antibiotics are used for the management of these mastitis an increase in the levels of endotoxins release can occur, depending on whether the action of the antibiotic is bactericidal or bacteriostatic, and an exacerbated inflammatory response with clinical consequences at the local and systemic level can occur [7]. Hence, treatments aimed at controlling this inflammatory reaction, specifically the deleterious levels of both macrophages and neutrophils and therefore of the cytokines that mediate inflammation, are also usually used in the management of Gram-negative associated mastitis [8].

There are different methods for the detection and quantification of endotoxins [9]. Among them, methods based on the use of horseshoe crab, *Limulus polyphemus*, amebocyte lysate (Limulus amebocyte lysate or LAL assays) are the most extended since their introduction in the 1960s [10]. There are currently three main methodologies based on LAL, all of them relying on the factor C coagulation cascade produced in the blood extract of horseshoe crabs after being exposed to endotoxins [1,9]. The gel-clot method is based on the gelling of the coagulogen protein in the presence of endotoxins and is a qualitative or semiquantitative assay. The chromogenic methods measure the colour developed by cleavage of a synthetic chromogenic substrate. Finally, the turbidimetric methods measure the turbidity caused by the precipitation of coagulogen protein. Both chromogenic and turbidimetric methods can be performed as kinetic assays by measuring colour or turbidity development throughout the incubation time and are quantitative assays. These methods are recognized in the pharmacopoeias of Europe, US, Japan, China, and other countries [11] for the detection of endotoxin activity in water and pharmacological products. In addition, there are guidelines at the European level for endotoxin detection in different matrices such as air (EN 14031:2021) or nanomaterials (ISO 29701:2010) using LAL assays. However, there are no standardized methodologies for the detection of endotoxin activity in milk although some researchers have used LAL based methodologies for this purpose [12,13,14,15,16,17,18].

The aim of this research is the optimization and validation of a kinetic turbidimetric technique based on LAL for the quantification of endotoxin activity in bovine milk.

## 2. Materials and Methods

### 2.1. Sample Collection

Bovine milk samples from different origins were collected to obtain a wide range of endotoxin activities. Briefly, 10 raw milk samples were from cows with clinical mastitis (showing swelling, redness, and/or oedema in one or more quarters) and were selected among the samples submitted for aetiological diagnosis of mastitis to Agrovet Laboratories (Mansilla Mayor, León, Spain). None of these animals was receiving any medication at the time of the sampling. Additionally, five bulk milk samples from different dairy farms and five commercial ultra-heat-treated (UHT) milk samples were tested. For all of them, a volume near to 5 ml of milk was aseptically collected in sterile tubes and kept at 4 °C immediately after collection. Upon arrival at the laboratory 1 ml of each sample was processed for conventional microbiological diagnosis (milk samples from mastitis affected cows) and somatic cells count (SCC) determination (Delaval Cell Counter DCC) (milk samples from mastitis affected cows and bulk milk samples) while the remaining volume was stored at −20 °C until processed for endotoxin determination.

### 2.2. Pre-processing of the Samples

All the materials used for the whole laboratory procedure, including plastic consumables, reagents, and water, were endotoxin-free and the entire process was carried out in a biological safety cabinet to avoid exogenous contamination by endotoxins.

Milk samples were thawed at room temperature. To eliminate possible interfering substances for analysis, 1 ml of each milk sample was filtered through a polyvinylidene fluoride (PVDF) filter of 0.45 µm pore size (Millex-HV, Merck-Millipore) using a sterile syringe. The filtrates were collected in sterile 1.5 ml microtubes and used to carry out serial dilutions using certified endotoxin-free ultra-pure water up to 1/1000 (1/10, 1/20, 1/100, 1/500, and 1/1000). When primary filtration was not possible due to the saturation of the filter, milk samples were pre-diluted 1/500 before being subjected to filtration.

### 2.3. Preparation of Endotoxin Standard Solutions

A commercial certified endotoxin standard (CSE *Escherichia coli*, Charles River) was resuspended in endotoxin-free ultra-pure water following the manufacturer’s instructions and used to prepare a standard solution with a concentration of 50 endotoxin units (EU) per mL. This standard solution was filtered in a similar manner to the milk samples before being used to prepare ten-fold dilutions containing 5, 0.5, and 0.05 EU/mL. These standard solutions were freshly prepared every day immediately before utilization and were used to construct a calibration curve for each procedure. 

### 2.4. LAL Assay

Endotoxin activity was quantified using a kinetic turbidimetric assay in which each sample is mixed with the LAL substrate reagent, placed in a turbidimetric reader at 37 °C and automatically monitored over the incubation. The time to the development of turbidity, usually refer as reaction time, is determined and is inversely proportional to the amount of endotoxin present in the sample. Finally, the concentration of endotoxin in the samples can be estimated using a standard curve. All the assays were performed in 96-well plates (96 well Endosafe-plates, Charles River, Barcelona, Spain) and using a commercial endotoxin detection kit (Kinetic turbidimetric LAL, Charles River), designed for endotoxin measurement in water and/or pharmaceutical products. Turbidity readings at 340 nm were carried out every 30 s during an incubation time of one hour (BIOTEK turbidimetric reader, Lonza, Porriño, Pontevedra, Spain).

Briefly, 100 µL of each standard solution (50 EU/mL, 5 EU/mL, 0.5 EU/mL, and 0.05 EU/mL), blank (endotoxin-free ultra-pure water) or diluted milk sample (1/500 and 1/1000) were dispensed per well. Two replicas of each standard, blank or sample were processed. Additionally, the potential interference of the matrix with the measurement was assessed by the addition of a known concentration of endotoxin and further estimation of the recovery ratio for each tested sample according to the manufacturer’s instructions. For this purpose, another two replicas of each diluted milk sample were tested by adding 100 µL per well and spiking with 10 µL of the standard solution containing 50 EU/mL. Finally, 100 µL of amebocyte lysate were added to each well and plates were incubated in the reader.

For each plate, a standard curve was generated by plotting the log of the average reaction time value for each standard solution against the log of endotoxin concentration and was used to estimate endotoxin activity in each sample.

The maximum allowed coefficient of variation (CV) between replicas of milk samples and standard solutions was established at 10% while the accepted recovery ratio of the spike was between 50 and 200%. 

### 2.5. Validation of the LAL Assay

Five measurements of each standard solution were made by two operators to validate the assay by determining its accuracy (precision/random error and trueness/bias) and uncertainty according to ISO 5725-1:1994 and ISO 19036:2019. A similar procedure was followed using a diluted negative milk sample spiked with endotoxin standard solution containing 50 EU/mL to provide milk samples containing 5, 0.5, and 0.05 EU/mL, as there is no commercial certified endotoxin standard for milk matrix.

Precision is defined as the closeness of the measurements made for the same sample and is usually estimated as the standard deviation (SD) while trueness describes the difference (absolute value) between the average concentration of endotoxin obtained in each determination and the corresponding theoretical concentration. Uncertainty was estimated to provide a quantitative indication of the reliability of a measurement. Finally, accuracy is the combination of trueness and precision, so the smaller the bias and the better the precision, the more accurate the measurement.

For the evaluation of the repeatability, four milk samples including two of medium endotoxin concentration (75–80 EU/mL), one of high endotoxin concentration (2000 EU/mL), and one of low concentration (28 EU/mL), were selected and repeated measurements were made by two operators according to ISO 19036:2019. Reproducibility was determined using 12 milk samples within each endotoxin concentrations (high, medium, and low) that were measured by two operators in different days and using two replicas per sample (ISO 19036:2019).

### 2.6. Statistical Analysis

The software IBM SPSS Statistics (version 26.0, Chicago, IL, USA) was used for statistical analysis. Endotoxin activity in milk from different sources was expressed as regular arithmetic mean (EU/mL) ± standard deviation as well as median and range. Correlation between endotoxin activity and SCC in Gram-positive or Gram-negative associated mastitis milk and bulk milk was evaluated using non-parametric Spearman’s rank correlation coefficient. 

## 3. Results

### 3.1. Effect of Filtration of the Samples

The effect of the pre-treatment of the samples by filtration through PVDF filter on endotoxin determination was evaluated using endotoxin standards and endotoxin-free water. As can be seen in the two examples shown in Figure 1, calibration curves obtained with filtered and unfiltered standards were very similar.

### 3.2. Calibration Curve, Coefficient of Variation and Recovery ratio

In all experiments, the linearity of the curve was very high within the range of concentrations tested and the correlation coefficient (R) for all the procedures was between 0.997 and 1.000. Likewise, the CV between replicas were in the range 0.00–3.22% for standards and blanks (Table 1) and 0.00–3.44% for milk samples. The recovery ratio in samples spiked with a known concentration of endotoxin was between 51 and 190% in more than 95% of the tested samples. 

### 3.3. Validation of the LAL Assay

Results obtained in the validation using five replicas of the endotoxin standards and spiked negative milk samples containing 5, 0.5 and 0.05 UI/mL measured by two different operators are shown in Table 2.

Results of the repeatability and reproducibility evaluation of the assay are shown in Table 3. Both parameters were greater in the medium range than in the high and low concentrations. 

### 3.4. Determination of Endotoxin Activity in Milk Samples

Endotoxin activity determined in milk samples from different sources is shown in Table 4. Among raw milk samples from cows with clinical mastitis results varied between <25 EU/mL and 29,000 EU/mL. The average endotoxin content was 19,642.5 ± 13,051.35 EU/mL (median 24,500.0, minimum 570.0, maximum 29,000.0) in milk samples from cows suffering from coliform mastitis (*n* = 4), while no endotoxin activity (<25 EU/mL) was reported in milk samples from clinically affected cows with mastitis associated to staphylococci (*n* = 3), *Streptococcus agalactiae* (*n* = 1), *Mycoplasma bovis* (*n* = 1), or *Serratia marcescens* (*n* = 1). Endotoxin activity varied between <25 EU/mL and 270 EU/mL (median <25) for bulk milk samples and between <25 and 175 EU/mL (median 57.08) for commercial UHT milk samples.

A significant positive correlation between endotoxin activity and SCC was demonstrated (Rho = 0.9, *p* = 0.037) for Gram-negative mastitis milk samples (Rho = 1.0, *p* < 0.01 for coliform mastitis milk samples), while no correlation was demonstrated for milk samples from Gram-positive mastitis or bulk milk samples (Rho = −0.147, *p* = 0.684).

## 4. Discussion

The release of endotoxins, mainly caused by bacterial lysis, is a major factor which can aggravate clinical disease in Gram-negative associated mastitis [5,6]. These endotoxin levels change with the progression of the disease but also with the antibiotic treatment which, depending on its mode of action, can significantly increase this release. Therefore, the identification of microorganisms involved and particularly the discrimination between Gram-positive and Gram-negative bacteria is crucial in the management of dairy cow mastitis, allowing for a tailored approach. Traditionally, diagnosis of Gram-negative associated mastitis has relied on clinical signs and microbiological culture results [13,14]. Among the firsts, inflammation of the quarter, pyrexia, anorexia, dehydration, and diarrhoea together with systemic endotoxemia are the main clinical signs reported in hyperacute presentations of coliform mastitis, while in acute cases, dehydration and diarrhoea do not usually occur [13]. However, diagnosis relying solely on clinical signs can be misleading and microbiological or molecular methods are needed for confirmation. Nevertheless, this diagnosis takes time and can delay the initiation of treatment. In this context, the detection of endotoxins in milk has been proposed as an alternative for the rapid identification of Gram-negative mastitis [14,18].

Although there are several methods for endotoxin detection, nowadays the Limulus amebocyte lysate (LAL) based tests are of choice due to their high sensitivity, potential for quantification as well as their easy performance [19]. These tests are safe, rapid, and theoretically would discriminate between Gram-negative and Gram-positive associated mastitis in few hours, allowing for the rapid initiation of an appropriate therapy. For this purpose, LAL tests can be performed on blood or milk, the latter being more appropriate for early diagnosis of mastitis as milk can contain endotoxin levels five times higher than those found in blood [13]. Several studies have reported the use of LAL conventional tests based on the macroscopic scoring of the coagulation for endotoxin detection in milk samples [12,13,14,16,17]. However, these assays are qualitative or semi-quantitative and are influenced by several environmental and laboratory conditions [20]. More recently, other authors have used chromogenic [15,18] or turbidimetric [21] LAL based tests adapted for the milk matrix. The aim of this research was to optimize and validate a kinetic turbidimetric LAL assay for the detection of endotoxins in bovine milk at an affordable cost (30–35 € per sample) which can be used in both research and routine diagnosis of bovine mastitis.

The colouration of the milk and the presence of interfering substances are among the main drawbacks for turbidimetric assays applied in milk samples. Dilution of the samples can be used to minimize the effect of both, colour and/or concentration of unwanted substances. An approach to calculate the maximum allowed dilution for endotoxin detection in pharmacological products, which depends on the posology and the route of administration of each product, has been proposed in the European Pharmacopoeia. Furthermore, a maximum allowed dilution has been suggested for endotoxin detection in sodium heparin using a gel-clot LAL assay [22]. However, there is no such proposal for milk matrix. A research project carried out in 2015 using a kinetic turbidimetric assay in milk from healthy cows used three different dilutions, 1/100, 1/200 and 1/400 [18]. Taking these data into account, we decided to use 1/500 and 1/1000 as working dilutions allowing for a detection range of 25–50,000 EU/mL, which ensures that the assay should be able to evaluate milk samples containing a wide range of endotoxins. 

In order to avoid interfering substances in the samples, a novel pre-processing of the milk samples by filtration was evaluated in our research. According to the European Pharmacopoeia, filtration is one of the most common methods for eliminating interference although some filters, usually those made of cellulose, may lead to false positive results due to cellulosic derivatives as glucans. In our study, filters with a pore size of 0.45 µm and made of polyvinylidene (PVDF) were used to filtrate all the tested milk samples. This material is highly chemically inert and is usually used in conditions that require high purity [23,24]. The effect of this filtration was evaluated using either endotoxin standards or endotoxin-free water as blank. The results obtained showed that this PVDF filtration does not affect the endotoxin concentration results. There was no false positive result and the variation in endotoxin activity measured in the standards was insignificant. Hence, filtration through PVDF filter was included as a routine pre-treatment of milk samples, standards, and blanks in our assay.

Moreover, the determination of the recovery ratio in milk samples spiked with a known amount of endotoxin allowed for the evaluation of interfering substances in the milk matrix as previously proposed [18]. According to the manufacturer of the commercial endotoxin detection kit, recovery rates must be between 50 and 200% in relation to the spiked concentration to be considered valid. More than 95% of the assays in the tested milk samples fulfilled this requirement, demonstrating the usefulness of the technique.

The turbidimetric LAL assay uses a calibration curve based on certified reference standards for bacterial endotoxins which allow for quantification. Linearity of the curves obtained (measured with the R^2^ coefficient) was very high, close to 0.99 in most of them, a fact which demonstrates the robustness of the method. To the authors’ knowledge there is no previous report on validation of turbidimetric LAL assays for the detection of endotoxins in milk samples. Accuracy, uncertainty, reproducibility, and repeatability were estimated following national (Spanish National Accreditation Entity, ENAC) and international (International Organization for Standardization, ISO) guidelines for validation. These parameters varied depending on the endotoxin activity of the samples. The best results were obtained in the medium range (0.5–5 EU/mL) while they get worse in the high (50 EU/mL) and low (0.05 EU/mL) concentration levels. 

The standardized assay was used on a selection of samples from different sources. It was able to clearly discriminate coliform mastitis, providing high endotoxin activities, and mastitis caused by Gram positive bacteria, with no endotoxin activity, confirming the results of a previous research using 12 milk samples from clinically affected cows [18]. Unexpectedly, no endotoxin activity was detected on a milk sample of a cow with clinical mastitis attributed to *Serratia marcescens*, a Gram-negative enterobacteria occasionally involved in dairy mastitis. This finding may be the consequence of a misdiagnosis but could also be due to a limited bacterial multiplication. In this sense, the fact that intramammary infections by *Serratia* spp. are usually associated with less severe clinical signs compared to other Gram-negative bacteria [25] could explain this unexpected result. Further studies are required to evaluate the usefulness of the assay in mastitis milk samples associated to Gram-negative bacteria other than *E. coli*.

The assay was also able to measure endotoxin activity in bulk milk and in commercial UHT milk samples as previously reported [11,17]. Gram-negative bacteria in raw milk are the main source of endotoxin in dairy products. Thermal treatments can effectively kill bacteria present in milk but do not remove endotoxin, which is a highly thermal stable molecule and allows for endotoxin activity in dairy products [26]. Hence, total bacterial counts in raw milk and intensity of thermal treatments are the main determinants of endotoxin activity in these dairy products [11]. Although there is no clear evidence that oral exposure to endotoxin can cause discomfort or disease, its association with severe immune response in humans and animals has been proved [11,26]. Methods to reduce endotoxin activity in dairy products are worth discussing and the standardized turbidimetric LAL assay can provide a useful tool for monitoring.

## 5. Conclusions

To sum up, our results demonstrate that the turbidimetric LAL assay developed is a robust and standardized method suitable for the detection of endotoxins in raw and commercial cow milk. Dilution and PVDF filtration of the samples avoid interferences associated to the milk matrix allowing the detection of a wide range of endotoxin activities. The assay can be used for the rapid diagnosis of coliform mastitis although further research is needed to evaluate its usefulness in other Gram-negative associated mastitis.

## Figures and Tables

**Figure 1 animals-13-00427-f001:**
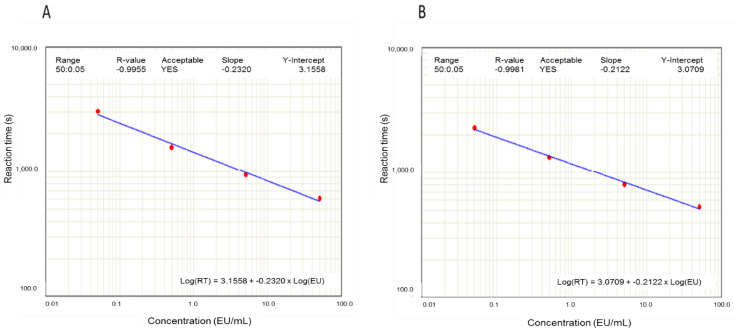
Calibration curves obtained with endotoxin standards (50, 5, 0.5, and 0.05 EU/mL). Effect of filtration: **A**: Unfiltered standards; **B**: Filtered standards. EU: endotoxin units.

**Table 1 animals-13-00427-t001:** Coefficient of variation (CV) among replicas of the filtered and unfiltered endotoxin standards (50, 5, 0.5, and 0.05 EU/mL) and blanks (endotoxin-free ultra-pure water).

Coefficient of Variation (%)				
	50 EU/mL	5 EU/mL	0.5 EU/mL	0.05 EU/mL
Unfiltered standard	2.16	1.86	1.36	1.38
Filtered standard	2.99	2.34	1.25	0.64
Unfiltered blank	0	0	0	0
Filtered blank	0	0	0	0

EU: endotoxin units.

**Table 2 animals-13-00427-t002:** Validation of the kinetic turbidimetric Limulus amebocyte lysate (LAL) assay for endotoxin quantification in milk samples using endotoxin standards and spiked milk samples: precision, trueness, and uncertainty.

Concentration (EU/mL)	Mean (EU/mL)	Precision (SD)	Trueness	Uncertainty (U) (k = 2.95%)	Accuracy (%)
Standards: 5 replicas					
0.05	0.051	0.004	0.001	0.009	1.620
0.5	0.540	0.025	0.040	0.094	8.044
5	5.281	0.405	0.281	0.986	5.618
50	54.652	6.562	4.652	16.088	9.305
Spiked negative milk simples: 5 replicas					
0.05	0.053	0.009	0.003	0.020	6.600
0.5	0.521	0.042	0.021	0.094	4.260
5	5.340	0.390	0.340	1.035	6.800
Global validation values					
0.05	0.052	0.007	0.002	0.015	4.110
0.5	0.530	0.035	0.030	0.093	6.052
5	5.310	0.388	0.310	0.994	6.209

EU: endotoxin units; SD: standard deviation; U = 2* √ Precision^2^ + Trueness^2^.

**Table 3 animals-13-00427-t003:** Validation of the kinetic turbidimetric Limulus amebocyte lysate (LAL) assay for endotoxin quantification in milk matrix using samples in the high, medium, and low concentration range: repeatability and reproducibility.

		Mean *	SD	CV (%)
Repeatability				
Endotoxin concentration (EU/mL)	High	5.193	0.936	18.022
	Medium	0.696	0.185	26.562
	Low	0.363	0.031	8.535
Reproducibility				
Endotoxin concentration (EU/mL)	High	4.86	0.355	7.297
	Medium	0.15	0.018	11.863
	Low	0.06	0.004	6.937

EU: endotoxin units; SD: Standard deviation; CV: Coefficient of variation; * Uncorrected values: dilution should be taken into consideration to estimate endotoxin concentration.

**Table 4 animals-13-00427-t004:** Endotoxin levels (EU/mL), somatic cell counts (SCC) and microbiological results in cow milk samples from different sources.

Source	Reference	Endotoxin (EU/mL)	SCC	Bacteriological Results
Cows with clinical mastitis	151677	570	951,650	*Escherichia coli*
	151678	29,000	1,457,000	*E. coli*
	151679	22,000	1,120,500	*E. coli*
	151680	<25	856,930	Coagulase-negative staphylococci
	151681	<25	958,670	*Serratia marcescens*
	151682	<25	657,000	Coagulase-negative staphylococci
	182179	<25	858,000	*Staphylococcus aureus*
	182180	<25	856,930	*Streptococcus agalactiae*
	182181	<25	1,587,600	*Mycoplasma bovis*
	182182	27,000	1,389,000	*E. coli*
Bulk milk	149927	140	331,000	-
	182406	<25	435,000	-
	196517	<25	172,000	-
	196518	<25	224,000	-
	206036	270	733,000	-
Commercial UHT milk	232807	34	-	-
	232808	175	-	-
	232809	57.08	-	-
	232810	<25	-	-
	232811	164.5	-	-

EU: endotoxin units; -: non-processed.

## Data Availability

The data presented in this study are available on request from the corresponding author. The data are not publicly available because they come from various sources and have been processed for the purposes of this study.

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
