# Peer review of "Intralaboratory Validation of a Kinetic Turbidimetric Assay Based on Limulus Amebocyte Lysate (LAL) for Assessing Endotoxin Activity in Cow Milk"

_animals, 2023, doi:10.3390/ani13030427_

Round 1
Reviewer 1 Report
There are some notes on the manuscript
- In line 39, the authors wrote LPS as a keyword, but they did not previously define this term.
- In line 95, the authors wrote the abbreviation UHT, but did not define this term beforehand.
- In lines 110 and 115 the authors did not specify which water is used for dilution. Whether distilled water or double distilled water is used or other types of water.
- In line 170, the authors wrote that the results are given as the mean ± standard deviation. At this point what do you mean by the term "mean", is it the LSM mean or an arithmetic mean or some other mean?
- In line 177, the authors wrote that no significant differences were observed between unfiltered standards and filtered standards. But how to explain the reaction time for filtered standards (B) faster than for unfiltered standards (A) as shown in Figure 1?
- Table 3 shows that the CV% for endotoxin concentration is high relative to repeatability. is there an explanation?
- In lines 245 to 247 the authors write that this method is easy, fast, and safe. But they did not mention how much it would cost for this method to become attractive.
- In lines 312-313 the authors mention that bacteria are inactivated by the thermal treatment process of the milk. But thermal treatment not only inactivates bacteria, it kills them.
- the references are Okay, but only reference 23 has no page number.
- I received the figure and tables twice, once in the text and the other time between the conclusion and references. Is there an explanation?
Author Response
- In line 39, the authors wrote LPS as a keyword, but they did not previously define this term.
Thanks for the comment. Lipopolysaccharide instead of LPS has been included in the Keywords (line 41).
- In line 95, the authors wrote the abbreviation UHT, but did not define this term beforehand.
Again, thanks for the comment. The definition of the abbreviation (ultra-heat-treated) has been included in line 97.
- In lines 110 and 115 the authors did not specify which water is used for dilution. Whether distilled water or double distilled water is used or other types of water.
Endotoxin ultra pure water was used for dilutions. This information has been included in the manuscript (lines 112, 117, 136, and 197 [Table 1]).
- In line 170, the authors wrote that the results are given as the mean ± standard deviation. At this point what do you mean by the term “mean”, is it the LSM mean or and arithmetic mean or some other mean?
The mean used was regular arithmetic mean. Statistical analysis in Material and Methods section has included some additional information for better understanding (lines 172-176).
“SPSS (version 26.0) was used for statistical analysis. Endotoxin activity in milk from different sources was expressed as regular arithmetic mean (EU/ml) ± standard deviation as well as median and range. Correlation between endotoxin activity and SCC in Gram-positive or Gram-negative associated mastitis milk and bulk milk was evaluated using non-parametric Spearman's rank correlation coefficient.”
- In line 177, the authors wrote that no significant differences were observed between unfiltered standards and filtered standards. But how to explain the reaction time for filtered standards (B) faster than for unfiltered standards (A) as shown in Figure 1?
Thanks for the comment. According to our experience, the slight differences observed in reaction times between the two calibration curves shown in figure 1 are just due to random variations and not a consequence of a systematic faster reaction in filtered standards. Reaction time is always slightly different in different assays and a new calibration curve is generated for every reaction but linear equations as well as the slopes and R values of both, non-filtered and filtered standards, are very similar. For better understanding we have mentioned that the two calibration curves shown in Figure 1 are just two examples and we have removed the reference to significant difference that could be confusing (lines 181-182).
“As can be seen in the two examples shown in Figure 1, calibration curves obtained with filtered and unfiltered standards were very similar.”
- Table 3 shows that the CV% for endotoxin concentrations is high relative to repeatability, is there an explanation?
Thanks again for the comment. Repeatability is affected mainly by analyst differences, i.e. pipetting, small bubbles that are imperceptible… In contrast, reproducibility is mainly affected by differences between days/assays or by the stability of the samples. Nevertheless, we do not consider that the CV% values reported are particularly high.
- In lines 245 to 247 the authors write that this method is easy, fast and safe. But they did not mention how much it would cost for this method to become attractive.
Thanks for the comment. We think that sensitivity, potential for quantification, easy performance, safety and rapidness are significant reasons for making attractive this assay, regardless to its economic cost. All together the advantages compensate if this cost is higher than that of traditional techniques. Nevertheless, the cost per sample is 32.50 (aprox.) and would depend on the number of samples processed. We consider that this price can be perfectly assumed, taking into account the advantages that the method can provide. We fully agree that the cost of analysis is a relevant information for potential users of these assays and according to the reviewer suggestion, this estimated cost has been included (lines 266-267).
“The aim of this research was to optimize and validate a kinetic turbidimetric LAL assay for the detection of endotoxins in bovine milk at an affordable cost (30-35 € per sample) which can be used in both research and routine diagnosis of bovine mastitis.”
- In lines 312-313 the authors mention that bacteria are inactivated by the thermal treatment process of the milk. But thermal treatment not only inactivated bacteria, it kills them.
According to the suggestion we have change the sentence to make it more clear (lines 326-332).
“Thermal treatments can effectively kill bacteria present in milk but do not remove endotoxin which is a highly thermal stable molecule and allow for endotoxin activity in dairy products [26]. Hence, total bacterial counts in raw milk and intensity of thermal treatments are the main determinants of endotoxin activity in these dairy products [11].”
- The references are Okay, but reference 23 has no page number.
Page numbers have been included. Thanks a lot for the comment.
- I received the figure and tables twice, once in the text and the other time between the conclusion and references. Is there and explanation?
According to the instructions for the authors, figures and tables must be included in the text but also at the end of the document. This is the reason why they are shown twice in the manuscript. Sorry for this inconvenience.
Reviewer 2 Report
Overall, a well-designed paper that brings to light a very interesting topic. From my point of view, there some minor changes that may be made to improve the overall quality of the paper.
The introduction section is appropriate, the importance of mastitis for the dairy sector is well emphasized on. The methodology of the experiments seems well designed and described.
It is well known that the identification of the pathogen agent plays a crucial role in the selection of the specific therapeutic protocol and of course, in the success of that treatment. Discharging between Gram+ and Gram- bacteria enables a tailored approach for each mastitis case. Thus, in my opinion, authors should emphasize a little more in this direction, respectively, the advantages that this method may bring for improving therapeutical protocols in dairy farms.
Also, since endotoxins are also present in UHT milk, I consider it would be interesting to read the opinion of the authors regarding the presence of these endotoxins in UHT milk, in the Discussions section. The conclusion section may also be improved further.
Author Response
- “It is well known that the identification of the pathogen agent plays a crucial role in the selection of the specific therapeutic protocol and of course, in the success of that treatment. Discharging between Gram + and Gram- bacteria enables a tailored approach for each mastitis case. Thus, in my opinion, authors should emphasize a little more in this direction, respectively, the advantages that this method may bring for improving therapeutical protocols in dairy farms”.
We agree with the reviewer that rapid discharging between Gram-positive and Gram-negative bacteria is a very interesting approach that would allowed for a tailored approach. With have tried to emphasize this fact in the Discussion section (lines 239-242).
“Therefore, the identification of microorganisms involved and particularly the discrimination between Gram-positive and Gram-negative bacteria is crucial in the management of dairy cow mastitis, allowing for a tailored approach.”
- Also, since endotoxins are also present in UHT milk, I consider it would be interesting to read the opinion of the authors regarding the presence of these endotoxins in UHT milk, in the Discussions section. The conclusion section may also be improved further.
We agree with the reviewer that the detection of endotoxins in commercial UHT milk samples and the potential use of the standardized assay for their detection is an interesting topic. We have included some information in the Discussion section (lines 326-336).
“Thermal treatments can effectively kill bacteria present in milk but do not remove endotoxin which is a highly thermal stable molecule and allow for endotoxin activity in dairy products [26]. Hence, total bacterial counts in raw milk and intensity of thermal treatments are the main determinants of endotoxin activity in these dairy products [11]. Although there is no clear evidence that oral exposition to endotoxin can cause discomfort or disease, its association with severe immune response in humans and animals has been proved [11,26]. Methods to reduce endotoxin activity in dairy products are worth discussing and the standardized turbidimetric LAL assay can provide a useful tool for monitoring.”
Conclusion has been improved (also suggested by reviewer #3) (lines 338-345).
“To sum up, our results demonstrate that the turbidimetric LAL assay developed is a robust and standardized method suitable for the detection of endotoxins in raw and commercial cow milk. Dilution and PVDF filtration of the samples avoid interferences associated to the milk matrix allowing the detection of a wide range of endotoxin activities. The assay can be used in the rapid diagnosis of coliform mastitis although further research is needed to evaluate its usefulness in other Gram-negative associated mastitis.”
Reviewer 3 Report
The authors propose the validation of a kinetic turbidimetric assy based on Limulus amebocyte lysate (LAL) for the quantification of endotoxins in milk samples. The topic is of interest to the field, the detection method is already in use, at least as a quantitative assay, but needed validation for bovine milk. The manuscript is well written and deserves publication.
Some minor issues
Line 72: “coagulen protein” please explain or correct
Line 125: Turbidity
Line 238-241: it should be stated that the detection of bacteria can be achieved also by NGS methods, e.g. 16S and Internal Transcribed Spacer (ITS) ribosomal RNA (rRNA) sequencing.
Author Response
- The main problem with this manuscript is that it shows the method to detect E. coli endotoxins, not Gram-negative bacteria endotoxins. The article presents the correlation between E. coli endotoxins and SCC only. The authors had only milk samples contaminated with two kinds of Gram-negative bacteria (E. coli and Gram-negative Serratia marcescens). Hypothetically the test should work for all- Gram-negative bacteria, but the article does not prove that. The abstract, discussion and conclusion should be rewritten.
A commercial endotoxin detection kit (Kinetic turbidimetric LAL, Charles River), designed for endotoxin measurement in water and/or pharmaceutical products, was used and adapted to the milk matrix in our study. The assay, theoretically, detects endotoxins from all Gram-negative bacteria, not only E. coli. However, as the reviewer mentioned, there is an incongruous result in a single mastitis milk sample (SCC 958,670 cells/mL) in which Serratia marcensens was isolated with no endotoxin detection. There are several potential explanations for this unexpected finding. As we mentioned in the Discussion section, “This finding may be the consequence of a misdiagnosis but could also be due to a limited bacterial multiplication. In this sense, the fact that intramammary infections by Serratia spp. are usually associated with less severe clinical signs compared to other Gram-negative bacteria [25] could explain this unexpected result” (lines 318-321). As suggested by the reviewer, we have included a mention to this discordant result or potential limitation in the Abstract, Discussion and Conclusion sections.
Abstract (lines 31-33):
“Further studies are required to evaluate the performance of the assay in mastitis milk samples associated to Gram-negative bacteria other than E. coli as well as during the clinical course of these Gram-negative mastitis or after their treatment with antibiotics.”
Discussion (lines 321-323):
“Further studies are required to evaluate the usefulness of the assay in mastitis milk samples associated to Gram-negative bacteria other than E. coli.”
Conclusion (lines 338-345):
“To sum up, our results demonstrate that the turbidimetric LAL assay developed is a robust and standardized method suitable for the detection of endotoxins in raw and commercial cow milk. Dilution and PVDF filtration of the samples avoid interferences associated to the milk matrix allowing the detection of a wide range of endotoxin activities. The assay can be used for the rapid diagnosis of coliform mastitis although further research is needed to evaluate its usefulness in other Gram-negative associated mastitis.”
- Please add the limit of detection or method range for real milk samples
The detection range of the assay for real mil simples has been included, as suggested, in the Discussion section (lines 279-280). The detection limit of the method depends on the working dilution employed and we have chosen 1/500 and 1/1000 as working dilutions. Taking these data into account, the detection limit 25 EU/ml-50.000 EU/ml. Several tests were made at 1/100 dilution (which could increase the detection limit up to 5 EU/ml) but this dilution was insufficient to avoid interfering substances for some milk samples. On the other hand, if required (i.e in a sample showing higher concentrations than 50,000 EU/ml), the upper limit can be increased, just by using a further dilution of the milk sample (i.e. 1/2000 or 2/5000…).
“Taking these data into account, we decide to used 1/500 and 1/1000 as working dilutions allowing for a detection range of 25-50,000 EU/ml which ensures that the assay should be able to evaluate milk samples containing a wide range of endotoxins.”
- Table 2 – add units (concentration)
Done.
- Line 223 – this information is in the “Statistical analysis” section. Please remove it.
Done.
- “Statistical analysis” – complete this section.
The Statistical analysis subheading has been completed as it was also suggested by reviewer #1 (lines 172-176).
“SPSS (version 26.0) was used for statistical analysis. Endotoxin activity in milk from different sources was expressed as regular arithmetic mean (EU/ml) ± standard deviation as well as median and range. Correlation between endotoxin activity and SCC in Gram-positive or Gram-negative associated mastitis milk and bulk milk was evaluated using non-parametric Spearman's rank correlation coefficient.”
- Serratia marcescens is a gram negative bacteria. Did you include the results for this bacterium in the calculation correlation between endotoxin activity and SCC? It should be. The correlation is not 1. Use limit of detection for Serratia marcescens.
As suggested by the reviewer a new correlation coefficient between endotoxin activity and SCC estimated including all Gram-negative associated mastitis milk samples has been included (lines 231-234). The correlation for coliform mastitis is shown in brackets.
“A significant positive correlation between endotoxin activity and SCC was demonstrated (Rho = 0.9, p=<0.037) for Gram-negative mastitis milk samples (Rho = 1.0, p<0.01 for coliform mastitis milk samples) while no correlation was demonstrated for milk samples from Gram-positive mastitis or bulk milk samples (Rho = -0.147, p = 0.684).”
Reviewer 4 Report
The Article aimed implementation and validation of a kinetic turbidimetric assay based on Limulus amebocyte lysate (LAL) for the quantification of endotoxins in milk samples. Method validation is quite well planned and carried out. Only four-point calibration curves and no working range of the method may be questionable. The main problem with this manuscript is that it shows the method to detect E.coli endotoxins, not Gram-negative bacteria endotoxins. The article presents the correlation between E.coli endotoxins and SCC only. The authors had only milk samples contaminated with two kinds of Gram-negative bacteria (E.coli and Gram-negative Serratia marcescens). Hypothetically the test should work for all Gram-negative bacteria, but the article does not prove that. The abstract, discussion and conclusion should be rewritten.
Please add the limit of detection or method range for real milk samples.
Table 2 – add units (concentration)
Line 223 – this information is in the “Statistical analysis” section. Please remove it.
“Statistical analysis” – please complete this section.
Serratia marcescens is gram negative bacteria. Did You include the results for this bacterium in the calculation correlation between endotoxin activity and SCC? It should be. The correlation is not 1. Use limit of detection for Serratia marcescens.
Author Response
- Line 72: “coagulen protein” please explain or correct.
Thanks for the comment. Coagulen has been changed by coagulogen (line 74).
- Line 125: Turbidity
Thanks for suggestion. Done (line 127).
- Line 238-241: it should be stated that the detection of bacteria can be achieved also by NGS methods, e.g. 16S and Internal Transcribed Spacer (ITS) ribosomal RNA (rRNA) sequencing.
Thanks. As suggested we have mentioned molecular methods, in general, as a tool for mastitis diagnosis (lines 248-252).
“However, diagnosis relying solely on clinical signs can be misleading and microbiological or molecular methods are needed for confirmation. Nevertheless, this diagnosis takes time and can delay the instauration of treatment. In this context, the detection of endotoxins in milk has been proposed as an alternative for the rapid identification of Gram-negative mastitis [14,18].”